# Carboxyl-Terminal Decoy Epitopes in the Capsid Protein of Porcine Circovirus Type 2 Are Immunogenicity-Enhancers That Elicit Predominantly Specific Antibodies in Non-Vaccinated Pigs

**DOI:** 10.3390/v14112373

**Published:** 2022-10-27

**Authors:** Ling-Chu Hung

**Affiliations:** 1Animal Drugs Inspection Branch, Animal Health Research Institute, Council of Agriculture, Executive Yuan, Miaoli County 35054, Taiwan; lchung@mail.nvri.gov.tw; Tel.: +88-637584811; 2Animal Health Research Institute, Council of Agriculture, Executive Yuan, New Taipei City 25158, Taiwan; 3Livestock Research Institute, Council of Agriculture, Executive Yuan, Tainan 71246, Taiwan

**Keywords:** monoclonal antibody, porcine circovirus type 2a, capsid protein, decoy epitope, open reading frame proteins, vaccination, IgG, immunity, swine herd

## Abstract

In the context of the carboxyl-terminus (C-terminus) of the capsid protein of porcine circovirus type 2a (PCV2a) and PCV2a vaccines, this study aimed to explore its unrevealing cryptic epitope and its relation to PCV2-infected herd immunity. To discover the C-terminus of the capsid protein of PCV2a, monoclonal antibodies (mAbs) were generated in this work. Two mAbs bound the two minimal linear epitopes (_229_PPLKP_233_ and _228_DPPLNP_233_ (or _229_PPLNP_233_)), which were located at the C-terminus of the capsid proteins of PCV2a and PCV2b, respectively. One mAb bound to the minimal linear epitope (_220_QFREFNLK_227_, peptide P82), but it neither bound the virus-like particle (VLP) of PCV2a nor produced positive staining in PCV2a-infected cells by immunofluorescence assay. Further, the residues 220–227 were not accessible on the surface of the VLP on the three-dimensional model, but the residues 228–231 extend toward the VLP exterior. Immunoassays were conducted in this study to screen anti-viral peptide-specific IgGs, which could differentiate vaccinated pigs from non-vaccinated ones. The data show two _220_QFREFNLKDPPLKP_233_-containing peptides had a significantly higher binding reactivity with sera from PCV2-infected pigs in the control group than with sera from the VLP-vaccine group, particularly seen in sera from swine aged 15 weeks to 24 weeks. However, the peptide P82 had not this phenomenon in that test. This study confirmed that C-terminal epitopes play an important role in PCV2-induced decoy of swine humoral immunity.

## 1. Introduction

Porcine circovirus type 2 (PCV2) is widely considered to be one of the most important swine pathogens [1], not because it is a small non-enveloped virus [2] but because it is highly resistant to environmental conditions [3] and it impacts pork production [1]. The clinical signs include dullness, progressive emaciation, dyspnea, jaundice, hepatomegaly, enteritis, reproductive failure, lymphoid depletion, lymphadenopathy, low growth performance, and other manifestations [4,5,6,7]. PCV2-associated diseases and PCV2-associated mortality manifested as severe swine herd problems [1,4,6,7], particularly it most commonly affects pigs of 2 to 3 and 5 months of age [8]. One of the most cost-effective strategies to control PCV2-associated diseases is the administration of PCV2a vaccines in piglets and/or sows, therefore there has been a rapid rise in the use of commercial PCV2a vaccines to reduce viremia and PCV2 infectious pressure in swine herds [9,10,11,12,13] over the past decade. This might have caused the global trend shift from PCV2a to PCV2b first [14,15,16,17], and then PCV2d became a major genotype worldwide [18,19,20,21,22,23].

PCV2 contains the closed circular single-stranded DNA [2] which involves 11 open reading frames (ORFs) [24]. ORF1 and ORF2 genes are the two major ORFs. ORF1 encodes for the non-structural replicase proteins *Rep* and *Rep*’ [24,25]. The main structural capsid protein is only encoded by ORF2 and has a molecular mass of 27.8–30 kDa [24,26]. Since capsid protein is not only a crucial antigen for inducing neutralizing antibodies but also for developing a protective immune response against PCV2 infection [9,27,28,29]. Therefore capsid epitopes have received much attention over the last two decades. Previous researchers had discovered its antigenic domains (residues 65–87, 113–139, and 193–207) by using anti-PCV2 swine polyclonal sera and PEPSCAN analysis [30], conformational epitopes (residues 47–84, 165–200, and 230–233) by using chimeric PCV1 and PCV2 [31], the conformational epitope (composed of the motif the residues 231–233 and 1–60 together) by using mAbs with neutralizing activity against both PCV1 and PCV2 [32], linear B-cell epitopes (residues 156–162, 175–192, 195–202, 231–233, and 228–234) by using synthetic peptides and mAbs [32,33], crucial residues (residues 59, 63, 83, 130, 133, 206, and 210) on the surface of capsid protein which are responsible for the differential reactivity of mAbs to different PCV2 strains [34], and decoy epitopes (residues 169–180) [35,36,37]. Several studies had attempted to apply these epitopes to develop vaccine antigen [38,39,40,41,42] or diagnostic reagent [31,33,34,36,37].

Although vaccination against PCV2 infection has been widely utilized for disease control and effectively reduces PCV2-associated diseases [11,13,43], low levels of PCV2 infection were still detected in vaccinated pigs under field conditions [13]. It was difficult to evaluate the immune response of the PCV2a vaccine in field farms since the prevalence of seropositive animals has grown steadily by vaccination or infection [11,12,13,44,45]. Therefore some researchers developed the *Rep*’ protein antibody detection method to distinguish natural infection with PCV2 from subunit vaccine immunization [46]. Some researchers found that high levels of anti-capsid peptide (residues 169–180) activity are associated with naturally PCV2-infected animals [35,47,48]. They indicated this decoy epitope (residues 169–180), which is exposed on the surface of the capsid subunit, but, in the structural context of the virus-like particle (VLP), this epitope is buried and inaccessible to antibody [35]. The fundamental characteristics of decoy epitopes are directing the humoral response away from protective epitopes and inducing non-neutralizing antibodies against these epitopes [35,49,50,51,52]. Therefore, the specific antibodies which against decoy epitopes or VLP, provide opportunities to devise diagnostic tests for monitoring the immunological effects of vaccination [47]. The other view is that modification of decoy epitopes and other epitopes of PCV2 in vaccines could improve immunogenicity for PCV2 control since the shifting prevalence of circulating PCV2 subtype strains, which could be increasingly inadequate for current vaccines.

The following critical amino acid substitutions (at residue 206, 210, 215, 232, and 234) in the C-terminus of the capsid protein of various PCV2 genotypes were noticed in molecular genetic analysis of PCV2 sequences [19,20]. Particularly, the extra amino acid (residue _234_K) was in the C-terminus of the capsid protein of PCV2d strains that might influence host immune response, then these strains might favor survival, virulence, and circulating among vaccinated swine herds. For this reason, immunorelevant epitopes of the C-terminus of the capsid protein of PCV2 should be studied. According to previous reports, the antigenic domain (residues193–207) [30], conformational epitopes (within residues 165–200, and 230–233 of PCV2a [31], and residues 231–233 of PCV2b [32]), linear B-cell epitopes (residues 195–202, and 231–233) [32] were found by epitope mapping with mAbs against PCV2 virion or recombinant capsid protein. Likewise, another study demonstrated that the chemical synthesis of the peptide C3 (residues 195–233) can mimic the C-terminus of the capsid protein of PCV2b, particularly the immunoreactivity of the peptide C3 with sera from PCV2-infected pigs [42]. These anti-C3 mAbs recognized two core motifs (P62 (_228_DPPLNP_233_), and P67 (_228_DPPLNPK_234_)), and each motif was located at the C-terminus of the capsid protein of PCV2b and PCV2d, respectively [33]. The residues (_225_NLKDPPLNP_233_) of the C-terminus of the capsid protein of PCV2b have been reported to be critical to VLP assembly, cell entry, and propagation [53]. However, there is little information about the connection between the C-terminus of the capsid protein of PCV2 and PCV2-infected herd immunity. Based on previous studies [30,31,32,33], there should be epitopes on the sequence of the capsid protein of PCV2a between residues 205 and 233. If the synthetic capsid peptide (residues 205–233) can mimic the epitopes of the native capsid protein, it could elicit antibodies that could bind to the native capsid protein.

Therefore, this paper presents a new approach to verify the chemical synthesis of the peptide can also mimic the native capsid protein of PCV2a. This study generated six mAbs against the C-terminus of the capsid protein of PCV2a (peptide P64, residues 205–233) and defined their minimal binding regions. This study also used these mAbs and other antibodies to label the capsid protein and other viral proteins to determine conformational epitopes in PCV2 virus-infected porcine kidney cells and VLP. Further, this study used the UCSF Chimera version to figure out the locations of binding residues of mAbs on the 3-D Model of the VLP of PCV2. Furthermore, more than 60 viral peptides (including capsid peptides and other ORF peptides) were used as coating antigens to screen anti-viral peptide IgGs that could significantly differentiate vaccinated piglets from the non-vaccinated. The relationship among the C-terminus of the capsid protein, epitopes, and their antibody responses in the PCV2-infected herd with or without vaccination was also discovered.

## 2. Materials and Methods

### 2.1. Design of Synthesized Peptides

The peptide P64 [33] contained the C-terminal sequence of the capsid protein of PCV2a (GenBank: AY754021.1) between residues 205 and 233. The peptide P64 was appended with an NH_2_-terminal (N-terminal) cysteine during synthesis, which was required for conjugation with the keyhole limpet hemocyanin (Thermo Scientific, Rockford, IL, USA). Other peptides were synthesized by Yao-Hong Biotechnology Inc. (New Taipei, Taiwan), as listed in Table 1–2. These peptides contained the sequence of the capsid protein of PCV between residues 205 and 233, overlapping ten-mer peptides and associated peptides (4-mer, 5-mer, 6-mer, 7-mer, 8-mer, 9-mer, 11-mer, 12-mer, 14-mer, 18-mer, 19-mer, and 24-mer). The capsid peptides of PCV2b (GenBank: AAC35331.1), PCV2d (GenBank: EU909686.1), PCV2b-1c (GenBank: AY484410.1), and PCV1 (GenBank: AGI99550.1) were used in this study. These truncated peptides were used for the mapping of linear epitopes for anti-P64 mAbs binding, as listed in Table 1.

Other ORF proteins or peptides [33,42] were tested for their immunoreactivities with antisera, which were from PCV2-unvaccinated conventional farrow-to-finish herd accompanied by this experimental PCV2-vaccinated piglets. The aforementioned proteins or peptides, including the ORF3 peptides of PCV2b (GenBank: AAC35332.1), the ORF6 protein of PCV2 (GenBank: AAG41231.1), the ORF7 protein of PCV2 (GenBank: AAC35318.1), the ORF8 protein of PCV2 (GenBank: AAC35314.1), the ORF9 protein of PCV2 (GenBank: AAC35335.1), the ORF10 protein of PCV2 (GenBank: AAC35337.1), and the ORF11 protein of PCV2 (GenBank: AAG41230.1), the ORF2 peptide of PCV1 (GenBank: ULM77948.1), the ORF2 peptide of PCV2 (GenBank: BAP81686.1), the ORF2 peptides of PCV2a (GenBank: AY754021.1), the ORF2 peptides of PCV2b (GenBank: AAC35331.1), and the ORF2 peptides of PCV2d (GenBank: EU909686.1), were listed in Table 2. The purity of all peptides was assessed by high-performance liquid chromatography (LC-10ATVP serial dual plunger pump, Shimadzu, USA) and tested for the correct mass by Waters Micromass ZQ™ 2000 LC Mass Spectrometer (Waters, Milford, MA, USA).

### 2.2. Generation of MAbs Which against the C-Terminus of the Capsid Peptide of PCV2a (Residues 205–233, P64)

Animal experiments in this study were performed following the current legislation on ethical and welfare recommendations. The experiments followed the standards of the Guide of the Care and Use of Laboratory Animals. All animal work was reviewed and approved by the Institutional Animal Care and Use Committee (IACUC) of the Animal Health Research Institute. The IACUC approval number A04005 was given to this study.

The method is essentially the same as that used by the author with some adjustments. More detail can be found in the previous paper [33]. Briefly, four five-week-old, female, BALB/cByJNarl mice were purchased from a specific pathogen-free (SPF) colony (the National Applied Research Laboratories, Taiwan). The mice were maintained in isolation rooms in filtertop cages. Mice were inoculated with 50 μg of immunogen (the peptide P64 conjugated with KLH) emulsified in Freund’s adjuvant (Sigma-Aldrich, St. Louis, MO, USA). Subsequently, mice were boosted fortnightly with the same amount of immunogen. Three days after the final booster, the spleens of the mice were harvested for hybridoma generation. Hybridomas were screened for the secretion of anti-P64-specific mAbs by indirect enzyme-linked immunosorbent assay (iELISA) using the peptide P64 as a coating antigen. That secondary antibody solution (Peroxidase-conjugated goat anti-mouse IgG (subclasses 1 + 2a + 2b + 3, Fcγ, Jackson ImmunoResearch, West Grove, PA, USA)) was used at 1:2500 dilution for hybridoma screening. The hybridomas that secreted anti-P64 mAbs were subcloned by limited dilution of the cells. Then, ascites containing mAbs were collected as previously proposed.

### 2.3. Isotyping of MAbs

The heavy chain and the light chain of the six mAbs secreted by each hybridoma were determined by using an SBA Clonotyping System/HRP (Southern Biotechnology, Birmingham, AL, USA). It was performed according to the previously published method [33]. Briefly, the MaxiSorp microtiter plate (Thermo Fisher Scientific, Inc., Waltham, MA, USA) was coated with the capture antibody (goat anti-mouse Ig was diluted to 5 μg/mL in 0.05-M bicarbonate buffer (Sigma-Aldrich, St. Louis, MO, USA)) at 4 °C overnight. After three washes with PBS containing 0.05% Tween 20 (PBST), the plate was blocked with the blocking buffer (PBST containing 5% casein hydrolysate) at 37 °C for 30 min. After washing, a volume of 100 μL of each culture supernatant of hybridoma was added sequentially, and the plate was incubated at 37 °C for 2 h. After washing, 100 μL of HRP labeled goat anti-mouse IgG1, -IgG2a, -IgG2b, -IgG3, -IgA, -IgM, -κ light chain, and -λ light chain were added to appropriate wells of the plate. PBST served as the background. The plate was incubated at 37 °C for 1 h and afterward washed with PBST. Then, 100 μL of 2,2′-azino-bis(3-ethylbenzothiazoline-6-sulfonic acid) diammonium salt (ABTS, Sigma-Aldrich, St. Louis, MO, USA) buffer was added to each well. After 30 min, a SpectraMax M5 microplate reader (Molecular Devices, San Jose, CA, USA) was used to measure the optical density (OD) of each well at 405 nm.

### 2.4. Peptide Array-Based Epitope Mapping

The peptide array-based epitope mapping was performed by iELISA using truncated peptides (as shown in Table 1) as coating antigens. Briefly, peptides contained the C-terminus of the capsid peptide between residues 205 and 233, associated 10-mer peptides, and truncated derivatives. Two commercial virus-like particles (VLP) of PCV2 vaccines (VLP-A (Porcilis^®^ PCV, Intervet International B.V., An Boxmeer, The Netherlands), and VLP-B (Ingelvac CircoFLEX^®^, Boehringer Ingelheim, St Joseph, MO, USA)) and one anti-PCV2 mAb 36A9 (Ingenasa, Madrid, Spain) were also used in this experiment. Maxisorp plates were coated with each peptide (5 μg/mL) or vaccine (15 μL/mL) in bicarbonate buffer and incubated at 4 °C overnight. After three washes with PBST, the plates were blocked with the blocking buffer at 37 °C for 30 min. After washing, the culture supernatant of hybridoma was added, and plates were again incubated at 37 °C for 2 h. The anti-peptide P64 mouse serum (at a 1:1000 dilution) was used as the positive control, and the anti-PCV2 mAb 36A9 served as the control. After rinsing three times with PBST, the secondary antibody solution was applied to the appropriate wells. Peroxidase-conjugated goat anti-mouse IgG (subclasses 1 + 2a + 2b + 3, Fcγ, Jackson ImmunoResearch, West Grove, PA, USA) was used at a 1:2500 dilution for binding to the mAbs. After 1 h, the plates were washed three times. The colorimetric reaction was developed by using the ABTS solution. Following incubation for 30 min, the OD was measured at 405 nm.

### 2.5. Immunofluorescence Assay

The immunofluorescence assay was performed as previously described [42]. The Porcine Circovirus Type 2 (PCV-2, T657 strain) FA substrate slides (Cat No. SLD-IFA-PCV2, Lot. P140313-001, VMRD, Pullman, WA, USA) were incubated with a 1:100 dilution of antiserum from experimentally peptide-immunized mice, a 1:100 dilution of mAb (ascites), and a 1:100 dilution of PCV2 convalescent-phase swine antiserum. After incubation at 37 °C for 1 h, slides were rinsed in PBS and then soaked for 15 min in PBS. The slides were then incubated at 37 °C with fluorescein isothiocyanate (FITC)-conjugated goat anti-mouse IgG (Fcγ) at a 1:100 dilution or FITC-conjugated goat anti-pig IgG (H + L) (all from Jackson ImmunoResearch, West Grove, PA, USA) at a 1:100 dilution. After 30 min of incubation, the slides were washed with PBS and then incubated with 4, 6-Diamidino-2-phenylindole, dihydrochloride (DAPI, AAT Bioquest, Sunnyvale, CA, USA) at a 1:2300 dilution in PBS for 15 min. The slides were mounted under 50% glycerol and observed with an Olympus BX51 fluorescence microscope and SPOT Flex camera (Diagnostic Instrument Inc., Model 15.2 64MP, Sterling Heights, MI, USA).

### 2.6. Generation of Structural Images for Capsid Protein of PCV2a

The three-dimensional (3-D) structures of the capsid protein were used as approaches to figure out the binding interaction of peptides and antibodies. The structural model of the capsid protein and the VLP of PCV2a coordinates were retrieved from the Protein Data Bank (PDB) entries for the capsid protein of PCV2a (PDB code: 3JCI) [54], and images were generated using UCSF Chimera version 1.14 from the Resource for Biocomputing, Visualization, and Informatics at the University of California, San Francisco, USA [55].

### 2.7. Archival Pig Sera Collected from the Piglets following Vaccination or Unvaccinated Controls

The previous study herd had PCV2 infection without getting the PCV2 vaccine in the conventional pig farm with the farrow-to-finish operation [45]. All pigs were notched in their ear with a unique identification number on the day of birth. They were weaned at 4 weeks of age and transferred into nursery houses. Then, those weaning piglets were moved to the fattening house at 8 weeks of age. This study was previously approved by the IACUC of the Livestock Research Institute (approval no. LRIIACUC100-33) and complied with all of the requirements and regulations for animal experimentation, following current regulations in Taiwan.

Two piglets of the same sex and similar body weight (close to the mean body weight of the entire newborn litter) were selected from each litter. This study involved 20 newborn piglets (10 male and 10 female) of TLRI Black Pig No.1 (TBP), delivered from 10 sows between January and March 2011. One piglet from each litter was injected intramuscularly with the VLP-B vaccine (Ingelvac CircoFLEX^®^, 1.0 mL, one dose at 3 weeks of age). The other piglet from each litter was injected intramuscularly with 2.0 mL of saline at 3 weeks of age and served as a control. Blood samples from each pig were collected at 3, 6, 9, 12, 15, 18, 21, and 24 weeks of age. Subsequently, sera separated from the aforementioned blood samples and stored at −20 °C. At last, there were only three piglets from the control group that died during this experiment. One pig died before 18 weeks of age, and two died before 21 weeks of age.

All sera (*n* = 153) had been examined with the immunofluorescence assay and polymerase chain reaction. The result showed that both vaccinated pigs and the controls had suffered from PCV2 infection [45] (as Appendix A).

### 2.8. Detection of the Specific Antibodies against the Capsid Peptides or Other ORF Peptides in Sera among Vaccinated Piglets and Control Ones

First, primary screening was used to select anti-viral peptide-specific antibodies, which could significantly differentiate vaccinated piglets from non-vaccinated ones. Six sera were selected from six animals (three litters) at three different ages (18, 21, and 24 weeks of age, respectively). Each group contained three samples from three individuals of different ages. Two VLPs, two inactive PCV2 viruses (VC (chimeric PCV1/2, Fostera™, Zoetis, Charles City, IA, USA), and VD (PCV2a virus, Circovac^®^, Merial, Saint-Priest, France)), and 56 peptides (capsid peptides and other ORFs peptides) were used as coating antigens in iELISA test.

Second, among vaccinated (*n* = 10) and control (*n* = 10) animals, the anti-viral peptide-specific antibodies were detected in the serum samples from pigs at different ages (3, 6, 9, 12, 15, 18, 21, and 24 weeks of age). Then, the N-terminus of the capsid peptide (C1), the middle region of the capsid peptide (C2), eight C-terminus of the capsid peptides (C3, P30, P47, P64, P97, P82, P106, and P107), VLP-B, and five non-capsid peptides (N1 (ORF3 peptide), N2 (ORF6 peptide), N3 (ORF9 peptide), P114 (ORF10 peptide), and P115 (ORF11 peptide)) were used as coating antigens in iELISA test.

They performed as previously described for pig anti-peptide ELISA [56], with the exception that the capsid peptides and other ORFs peptides (as Table 2) were used as the coating antigen, and pig sera at a 1:200 dilution were used. The subsequent procedures of blocking, archival pig sera, washing, secondary antibody, substrate, and the OD reading, were as previously described [56].

### 2.9. Statistical Analysis

The data were analyzed by using a one-way analysis of variance (ANOVA) and Tukey’s Studentized Range (HSD, honestly significant difference) multiple comparisons test using the SAS Enterprise Guide 7.1^®^ software (SAS Institute Inc., Cary, NC, USA). A *p*-value < 0.05 was considered significant.

## 3. Results

### 3.1. Hybridomas Screening and Isotyping of MAbs against the C-Terminus of the Capsid Peptide of PCV2a (Peptide P64)

Following the fusions, hybridoma supernatants were screened for the presence of anti-capsid peptide (P64) specific antibodies by iELISA. For choosing hybridomas that only produce IgG mAb, the peroxidase-conjugated goat anti-mouse IgG (subclasses 1 + 2a + 2b + 3, Fcγ) secondary antibody was used for hybridoma screening. Among them, 41 hybridomas supernatants reacted with the peptide P64 at the first screening. After repeated subcloning by limiting dilution and selection, six stable hybridomas secreting anti-P64 mAbs were obtained. Three hybridomas (clone 4C9, 4F6, and 4H9) produced IgG1 mAb, two hybridomas (clone 6E3 and 6H11) produced IgG2b mAb, and one hybridoma (clone 3H11) produced IgG2a mAb. Those hybridomas produced mAbs with kappa-light chains. The results are presented in Table 3.

### 3.2. Mapping of Linear Epitopes for Anti-Capsid Peptide (P64) MAbs Binding

The peptide array-based epitope mapping was performed to determine the binding site of each mAb (Figure 1). The mAb 3H11 bound to 15 linear peptides (peptides P30, P47, P51, P59, P62, P64, P65, P71, P73, P84, P97, P98, P106, P107, and P108) and two VLPs of PCV2 (VLP-A (Porcilis^®^ PCV) and VLP-B (Ingelvac CircoFLEX^®^)). Notably, some peptides (P47, P64, P65, P71, P73, P84, P97, P98, P106, and P108) contained the common core peptide (P84 and _229_PPLKP_233_), which was located at the C-terminus of the capsid proteins of PCV2a. Others (peptides P30, P51, P59, P62, and P107) contained the common core peptide (P62 and _228_DPPLNP_233_), which were located at the C-terminus of the capsid proteins of PCV2b. Therefore, the mAb 3H11 bound the two minimal linear epitopes (PPLKP and DPPL**N**P). Similarly, the mAb 6H11 is bound to 16 linear peptides and two VLPs of PCV2. It bound the two minimal linear epitopes (_229_PPLKP_233_ and _229_PPL**N**P_233_), which were located at the C-terminus of the capsid proteins of PCV2a and PCV2b, respectively. Interestingly, the mAb 4F6 bound to 9 linear peptides (peptides P39, P64, P79, P80, P82, P83, P97, P106, P107, and P108), but it could not react with all VLPs of PCV2. The mAb 4F6 reacted with these peptides that contained the common core peptide (P88 and _220_QFREFNL_226_). However, the mAb 4F6 did not bind firmly to the peptide P88. Otherwise, the mAb 4F6 bound strongly to the minimal linear epitope (peptide P82 and _220_QFREFNLK_227_). The binding interaction of mAb 4F6 and peptide P82 was stronger than that of the mAb 4F6 and other linear peptides (P39, P79, P80, P83, P107, and P108). Two mAbs (4C9 and 4H9) not only reacted with the peptide of immunogen (peptide P64) but also reacted mildly with the peptide P108 (_210_DYNIRVTMYVQFREFNLKDPPLKP_233_). These data indicated that the mAb 6E3 reacted only with the peptide of immunogen (P64). The positive control (mAb 36A9) not only reacted with VLPs of PCV2 but also showed minor reactions against some linear peptides (P64, P86, and P108).

### 3.3. Immunoreactivity of Anti-P64 MAbs with PCV2 Virus

The specificity of mouse immune serum, pig sera, and mAbs was primarily determined by IFA with the PCV2 FA substrate slides according to the manufacturer’s manual. Slides were PCV2 virus (strain: T657)-infected porcine kidney cells fixed on the surface of Teflon-masked slides. The data showed that the PCV2-infected pig antiserum produced strong intranuclear and cytoplasmic staining in PCV2-infected PK cells (Figure 2C, D). This staining was used to identify all viral proteins of PCV2. When PCV2 FA substrate slides were stained with anti-P64 mouse sera, positive signals were displayed in a local distribution in the nucleus only (Figure 2E,F). This step was used to identify the C-terminus of the capsid protein of PCV2a. In the same way, the anti-VLP of PCV2 mouse sera produced dispersing granular and cytoplasmic staining in PCV2-infected PK cells (Figure 2G,H). These findings were similar to the previous study [34]. The previous data indicated that the anti-peptide C3 (PCV2b) mAbs could not specifically detect this PCV2 virus (strain: T657) [34]. Surprisingly, this study showed that not only the mAb 3H11 produced strong specific intranuclear staining and slightly cytoplasmic staining in PCV2-infected PK cells (Figure 2I,J), but also the mAb 6H11 produced strong specific intranuclear staining in PCV2-infected PK cells (Figure 2S,T). Based on these data, this strain (T657) was assumed to belong to PCV2a. Inexplicably, the mAb 4C9 produced unspecific cytoplasmic staining in PCV2-infected PK cells (Figure 2K,L), and the mAb 4H9 produced intranuclear staining and slightly unspecific cytoplasmic staining in PCV2-infected PK cells (Figure 2O,P). The mAbs (4F6 and 6E3) did not recognize this PCV2 virus (strain: T657) in PCV2-infected PK cells (Figure 2M,N,Q,R). The positive signals were due to different abundance of the protein, the degradation of the protein, different PCV2 strains, conformational epitopes, or linear epitopes, which interact with the antibody.

### 3.4. Locations of Binding Residues of MAbs (3H11, 4F6, and 6H11) on the 3-D Model of the VLP of PCV2

The crystal structure of capsid proteins (residues 42–231) of PCV2a [Protein Data Bank (PDB) with accession code 3JCI [54] was reported and Chimera software was used to spatially visualize the locations of critical binding residues of mAbs (3H11, 4F6, and 6H11). Both of the binding residues (residues 228–231) of mAbs (3H11 and 6H11) and the binding residues (residues 220–227) of mAb 4F6 were mapped onto the 3-D structure of VLP of PCV2a and a single capsid protein subunit. The residues (_228_DPPL_231_) were highlighted in red, residues (_220_QFREFNLK_227_) were labeled in yellow, and other residues (_205_SKYDQDYNIRVTMYV_219_) were labeled in blue (Figure 3A–F). These aforementioned color residues represented the peptide of immunogen (P64) that was depicted on the ribbon and surface maps of a single capsid protein subunit. The critical binding residues of mAbs (3H11 and 6H11) extend toward the VLP exterior (Figure 3E,F). However, the binding residues of mAb 4F6 were not accessible on the surface of the VLP (Figure 3E,F), excluding two residues (_225_N and _227_K). Although these two residues (_225_N and _227_K) were obviously on the surface of a single capsid protein subunit (Figure 3C), they were almost buried after the self-assembling of capsid proteins into a VLP (Figure 3E,F).

### 3.5. Immunoreactivities of Peptides with Swine Sera Which Were from Vaccinated Piglets or Control Ones

For primary screening, there was no significant difference between the vaccine group and the control group in non-capsid or non-C-terminus of the capsid peptides ELISA tests (Figure 4A). However, the striking result to emerge from the data was that significantly different interactive affinity between these two groups in the same C-terminus of the capsid peptides ELISA (Figure 4B) and the VLP-A ELISA tests(Figure 4A).

C-terminal capsid peptides (P30, P47, P65, P71, P73, P84, P97, P98, P106, and P108) had a significantly higher binding interaction with pig sera from the control group than with sera from the vaccine group (Figure 4B). It is worth noting that most of the aforementioned peptides belong to PCV2a. In another way, the VLP-A had a significantly higher binding interaction with pig sera from the vaccine group than with sera from the control group. Although Trible et al. indicated the peptide (_169_STIDYFQPNNKR_180_, P93) was an immunological decoy that provides the basis for diagnostic approaches that can differentiate infected from vaccinated animals [36,47,48], it was unlikely in this result (Figure 4A). Both control and vaccinated animals had suffered PCV2 infection (Appendix A). Remarkably, the capsid peptides (P101, _132_KATALTYDPYVNYSS_146_, and P103, _117_GVGSSAVILDDNFVT_131_) had higher mean OD values of binding interaction with pig sera from the control group than with sera from the vaccine group, but their data did not show a significant difference in this experiment since several potential shortcomings needed to be considered in this primary screening test.

Previous studies demonstrated that each anti-viral protein IgG had a different curve of the profile in the same herd after vaccination [56]. To verify C-terminus of the capsid peptides can differentiate vaccinated from control animals, anti-viral peptide IgG in archival sera from pigs at different age were used in this study. Some C-terminus of the capsid peptides (C3, P47, P64, P97, P106, and P107) had a significantly higher binding interaction with pig sera from the control group than with sera from the vaccine group aged 15 weeks to 24 weeks (Appendix A and Figure 5B–F). The peptide P30 had a similar effect on pig sera aged 18 weeks to 24 weeks (Figure 5A). Significantly, using PCV2a peptides (P47 or P106) reacted with pig sera, which had a larger difference in OD value between the control group and vaccine group than that using the same length of PCV2b peptides (P30 or P107), particularly swine aged 15 weeks to 24 weeks. Furthermore, two _220_QFREFNLKDPPLKP_233_-containing peptides (P97 or P106) had better results of this phenomenon than did other _225_NLKDPPLKP_233_ (or _225_NLKDPPL**N**P_233_)-containing peptides (C3, P30, P47, P64, and P107). These peptides can be used as a tool to differentiate vaccinated from non-vaccinated animals which had suffered PCV2 infection. The cut-off value of anti-P97 IgG and anti-P106 IgG ELISA was 1.9 and 1.4, respectively.

As anticipated, this experiment showed the immunogen VLP-B had a significantly higher binding interaction with pig sera from the vaccine group than with sera from the control group whose age was 9 weeks to 15 weeks (Appendix A). The middle region of the capsid peptide (C2) and ORF3 peptide (N1) had a significantly higher binding interaction with pig sera from the control group than with sera from the vaccine group at 24 weeks and 15 weeks of age, respectively (Appendix A). The N-terminus of the capsid peptide (C1), ORF6 peptide (N2), ORF9 peptide (N3), the binding residues (P82) of mAb 4F6, ORF10 peptide (P114), and ORF11 peptide (P115) could not be used as a tool to differentiate vaccinated from non-vaccinated animals that had suffered PCV2 infection (Appendix A).

## 4. Discussion

According to previous work, the chemical synthesis of peptide C3 (residues 195–233), can mimic the native form of the C-terminus of the capsid protein of PCV2b [42] and its epitopes [33]. The peptide P64 (residues 205–233) of the capsid protein of PCV2a was synthesized and used in this study. These experiments corroborate previous results. It is not only the anti-P64 mouse sera that recognized the native form of viral protein in PCV2-infected PK cells by IFA but also the peptide P64 showed good immunoreactivity with sera from PCV2-infected pigs. Subsequently, anti-P64 mAbs were generated and characterized in this study. This study showed that anti-P64 mAb (3H11 or 6H11) bound the two minimal linear epitopes (_229_PPLKP_233_ and _228_DPPLNP_233_ (or _229_PPLNP_233_)), which were located at the C-terminus of the capsid proteins of PCV2a and PCV2b, respectively. This phenomenon lends support to previous findings about two types of C-terminus of the capsid protein of PCV2 (PCV2b and PCV2d) that were recognized by anti-C3 mAbs with pluripotency of binding [33]. The single most conspicuous observation to emerge from these phenomenon comparisons was C-terminus of the capsid protein of PCV2a could elicit the specific antibody against both PCV2a and PCV2b, then PCV2b one could elicit the specific antibody against both PCV2b and PCV2d. It is very similar to the previous study reports that widely used PCV2a-based vaccines lead to a decrease in the prevalence and viremia level of PCV2a and PCV2b in field pig serum collected from 12 states in the United States in 2012 compared to data from the pre-vaccination period of 2006 [57]. The shift in circulating genotype prevalence to PCV2d may reflect the lack of available commercial PCV2b-based vaccines, which warrants further study for confirmation. Although some studies demonstrated that the PCV2a-based vaccine was effective in reducing PCV2d viremia [58], their data showed that serum viremia and shedding of the virus were still observed at 3 weeks or 6 weeks after challenge in vaccinated pigs [58,59].

The data showed the binding interaction of anti-P64 mAb 6H11 and P85 (OD value: 0.22 ± 0.02) was weaker than that of mAb 6H11 and P84 (OD value: 0.92 ± 0.05). The P85 (_230_PLKP_233_) should be the minimal linear epitope, although the mAb 3H11 did not react with P85. However, the mAb 3H11 reacted with P84 (_229_PPLKP_233_) and recognized the PCV2 virus (strain: T657) in PCV2-infected PK cells. This is in good agreement with previously reported conformational epitopes within residues 230–233 of PCV2a [31]. According to the crystal structure of capsid proteins (residues 42–231) of PCV2a, this study showed that the residues _228_DPPL_231_ extend toward the VLP exterior. The residues _232_KP_233_ were not shown in this crystal structure (PDB code: 3JCI) [54], but _232_KP_233_ was supposed to extend toward the VLP exterior (Figure 3E–F). The data also showed that the binding interaction of mAb 3H11 (or 6H11) and P65 (_228_DPPLKP_233_) was stronger than that of the mAb 3H11 (or 6H11) and P84 (Figure 1). It is expected that the antigenicity of P65 (_228_DPPLKP_233_) was better than P84 (_229_PPLKP_233_) in swine immune response, but the truth was: not yet (Figure 4B). This study also revealed that the mAb 4F6 bound strongly to the minimal linear epitope (P82 and _220_QFREFNLK_227_), but it neither bound the VLP of PCV2a nor recognized native viral protein in PCV2-infected PK cells by IFA. Although residues in this epitope (P82 and _220_QFREFNLK_227_) are located on the interior surface of a single capsid protein subunit (Appendix A), major residues (P89 and _220_QFREFN_225_) of this epitope are completely buried in the VLP (Figure 3E–F). This epitope (P82 and _220_QFREFNLK_227_) showed mild immunoreactivities (OD value: 0.59 ± 0.04) with sera from PCV2-infected pigs at 24 weeks of age (Appendix A).

It has been suggested [35,36,60] that an immune response against the interior (or decoy) epitope on the disassembled monomer or fragments of capsid protein is produced during virus replication. They also demonstrated that swine immunity following VLP vaccination favored PCV2 neutralizing activity; whereas PCV2 infection and disease produced high levels of non-neutralizing antibody, primarily directed against that decoy epitope (residues 169–180)-containing polypeptide (residues 160–233 of capsid protein) [36] or ubiquitin protein (residues 43–233 of capsid protein) [35]. Therefore, they indicated that the level of anti-decoy epitope (residues 169–180) antibody can differentiate infected from vaccinated animals (DIVA) [35,36,47,61]. According to the aforementioned hypothesis and results, the interior epitope (P82 and _220_QFREFNLK_227_) might be an unrevealing decoy epitope. However, the data showed that interior epitope (P82 and _220_QFREFNLK_227_) could not be used as the tool to differentiate vaccinated from non-vaccinated animals that had suffered PCV2 infection. The most striking observation to emerge from the data comparison was that _229_PPLKP_233_-containing peptides had a significantly higher binding interaction with pig sera from non-vaccinated animals than with sera from VLP-vaccinated animals. Particularly, _227_KDPPLKP_233_-containing peptides or _220_QFREFNLKDPPLKP_233_-containing peptides (P97 or P106) did that test which had a larger difference in OD value between the control group and vaccine group whose ages 15 weeks to 24 weeks. Those residues are just local at the C-terminal epitope (_229_PPLKP_233_) and the interior epitope (P82 and _220_QFREFNLK_227_). A previous study indicated that the C-terminus of the capsid protein of PCV2 is critical to VLP assembly, cell entry, and propagation [53]. Therefore, C-terminal epitopes should play an important role in PCV2-infected swine herds.

In general, these results confirm the conformational epitope (_229_PPLKP_233_) and interior epitope (_220_QFREFNLK_227_) of the C-terminus of the capsid proteins of PCV2a. This study revealed that the _229_PPLKP_233_-containing peptides had a significantly higher binding interaction with pig sera from non-vaccinated animals than those sera from VLP-vaccinated animals. Particularly, _220_QFREFNLKDPPLKP_233_-containing peptides (P97 or P106) did that test. It may be assumed that PCV2a avoids swine immune response against this important conformational epitope (_229_PPLKP_233_ or _230_PLKP_233_) by eliciting anti-decoy epitope-specific antibodies which do not recognize this conformational epitope (_229_PPLKP_233_ or _230_PLKP_233_) of the C-terminus. For this reason, those decoy peptides should be connected to this conformational epitope (_229_PPLKP_233_) of the C-terminus to elicit specific antibodies to replace anti-_229_PPLKP_233_ antibody production. It is recommended that further research should be undertaken on this interaction of host and virus. At last, this approach has the potential to be used as a tool to differentiate VLP-vaccinated from non-vaccinated animals that had suffered PCV2 infection in swine herds, but the major spreading PCV2 strains in the field must be taken into account.

## 5. Conclusions

To sum up, this work generated anti-P64 mAbs against the C-terminus of the capsid peptide (residues 205–233) of PCV2a. This study confirmed the mAb (3H11 or 6H11) bound the two minimal linear epitopes (_229_PPLKP_233_ and _228_DPPLNP_233_ (or _229_PPLNP_233_)), which were located at the C-terminus of the capsid proteins of PCV2a and PCV2b, respectively. Both mAb 3H11 and 6H11 produced positive staining in PCV2-infected PK cells by IFA. The mAb 4F6 bound strongly to the minimal linear epitope (_220_QFREFNLK_227_), but it neither bound the VLP of PCV2a nor produced positive staining in PCV2-infected PK cells. These data showed that the residues of 229–233-containing capsid peptides of PCV2a had a significantly higher binding interaction with pig sera from the control group than with sera from the VLP vaccine group in primary screening. Furthermore, two _220_QFREFNLKDPPLKP_233_-containing peptides (P97 or P106) had better results of this phenomenon than did other residues of 225–233-containing peptides, particularly swine aged 15 weeks to 24 weeks. These peptides can be used as a tool to differentiate VLP-vaccinated from non-vaccinated animals which had suffered PCV2 infection. In the structural view, residues 228–231 extend toward the VLP exterior, otherwise, residues 220–227 were not accessible on the surface of the VLP on the 3-D Model. This protruding epitope cooperates with the buried epitope to elicit specific IgG in PCV2-infected pigs, particularly which had not been inoculated with the VLP-B vaccine, but the sample size needs to be considered.

## Figures and Tables

**Figure 1 viruses-14-02373-f001:**
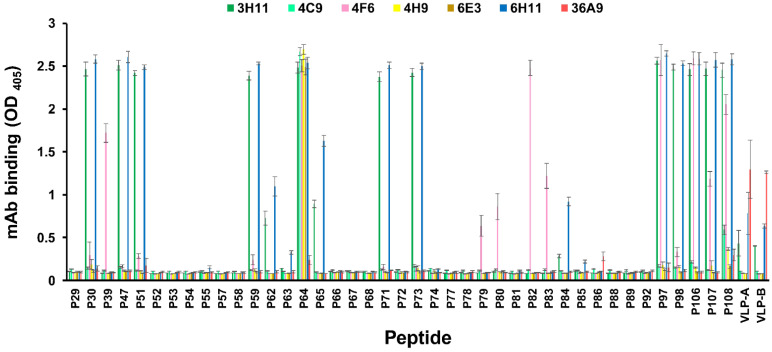
Mapping of the linear and minimal epitopes for anti-capsid peptide (peptide P64) mAbs binding. Anti-P64 mAbs bound the linear peptide spanning from residues 205 to 233. The mAb binding was tested by using an iELISA. Peptides contained the sequence of the capsid protein between residues 205 and 233, associated 10-mer peptides, and truncated derivatives (as shown in Table 1). The experiments were performed three times. Data represent the mean ± standard error (SE).

**Figure 2 viruses-14-02373-f002:**
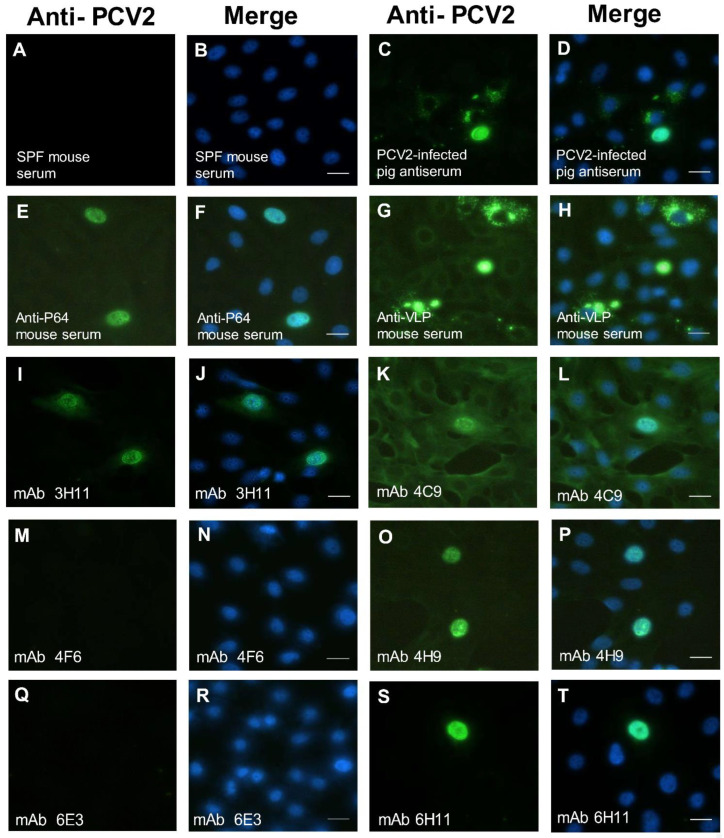
The localization of viral proteins of PCV2 on PCV2 (strain: T657)-infected porcine kidney cells. The localization of viral proteins of PCV2 was assessed by indirect immunofluorescence assay (IFA) using anti-PCV2 polyclonal antisera (**C**–**H**) and anti-P64 mAbs (**I**–**T**) on the Porcine Circovirus Type 2 FA substrate slide (VMRD). Staining with SPF mouse serum was used as the negative control (**A**,**B**). Staining with PCV2-infected pig antiserum (**C**,**D**), anti-P64 mouse serum (**E**,**F**), and anti-VLP mouse serum (**G**,**H**) were used as positive controls and to identify all viral proteins of PCV2 (**C**,**D**), C-terminus of the capsid protein of PCV2a (**E**,**F**), and VLP of PCV2a (**G**,**H**), respectively. Left column (**A**,**C**,**E**,**G**,**I**,**K**,**M**,**O**,**Q**,**S**): Fluorescence microscopy of viral proteins of PCV2 was identified (green). Right column (**B**,**D**,**F**,**H**,**J**,**L**,**N**,**P**,**R**,**T**): Nuclei were stained with DAPI (blue) and the merge of the images was shown. The scale bar is 20 μm.

**Figure 3 viruses-14-02373-f003:**
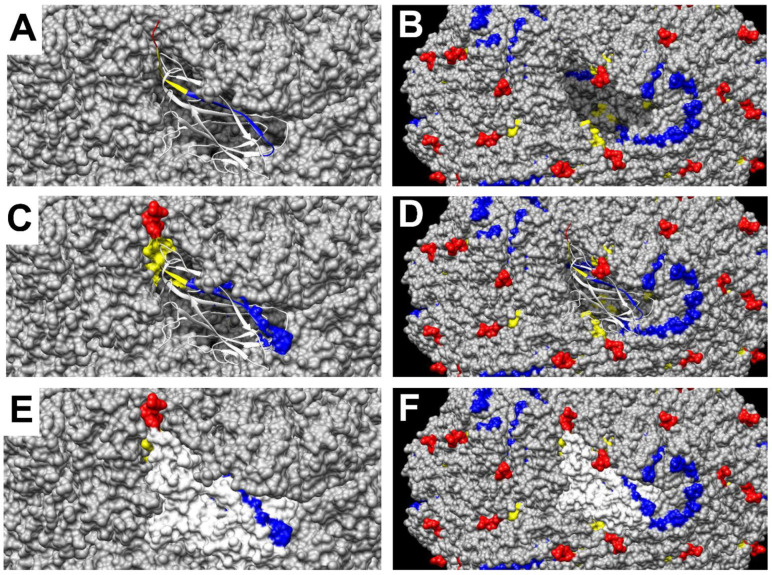
Location of amino acid residues of designed peptide (P64) and critical binding residues on the capsid protein of PCV2a. (**A**) The secondary structures of the single capsid protein of PCV2a were represented as a ribbon diagram in one VLP of PCV2a. The residues 228–231 were highlighted in red, the residues 220–227 were labeled in yellow, and other residues 205–219 were labeled in blue (**A**–**F**). These color residues represented the peptide P64. (**B**) The VLP of PCV2a lacks a single capsid protein subunit. (**C**) 3-D model of the peptide P64 rendered as a solid surface. (**D**) The secondary structure of the single capsid protein is represented as a ribbon diagram in one VLP. (**E**) 3-D model of the single capsid protein subunit rendered as a solid surface. (**F**) The surface map of all capsid proteins subunits in one VLP. The structural model of the capsid protein and the VLP of PCV2a coordinates were retrieved from the Protein Data Bank (PDB) entries for the capsid protein of PCV2a (PDB code: 3JCI) [54], and images were generated using UCSF Chimera version 1.14 from the Resource for Biocomputing, Visualization, and Informatics at the University of California, San Francisco, USA [55].

**Figure 4 viruses-14-02373-f004:**
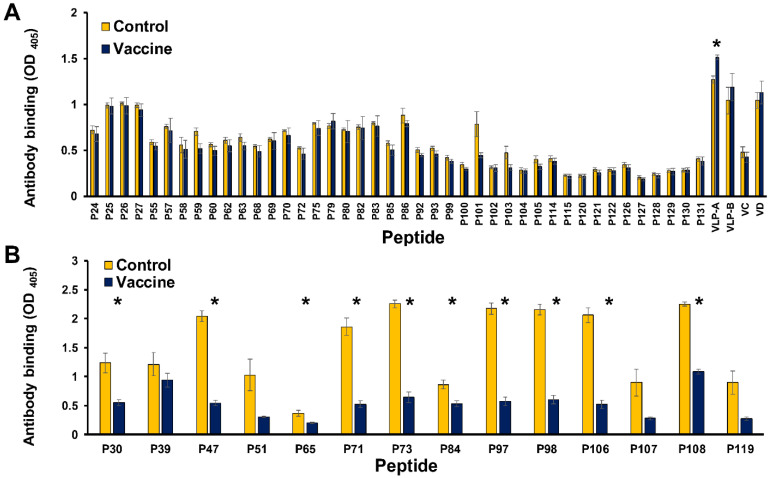
Screening anti-viral peptide-specific IgGs could differentiate PCV2-vaccinated pigs from non-vaccinated ones. Each group contained three samples from three individuals of different ages (18, 21, and 24 weeks of age, respectively). (**A**) 42 peptides (capsid peptides and other ORFs peptides), two VLPs (VLP-A, VLP-B), and two inactive PCV2 viruses (VC, VD) were used as coating antigens in the iELISA test. (**B**) 14 peptides (C-terminus of the capsid peptides) were used as coating antigens in the iELISA test. The data represent the mean ± standard error. Significant *p* values are indicated as * *p* < 0.05.

**Figure 5 viruses-14-02373-f005:**
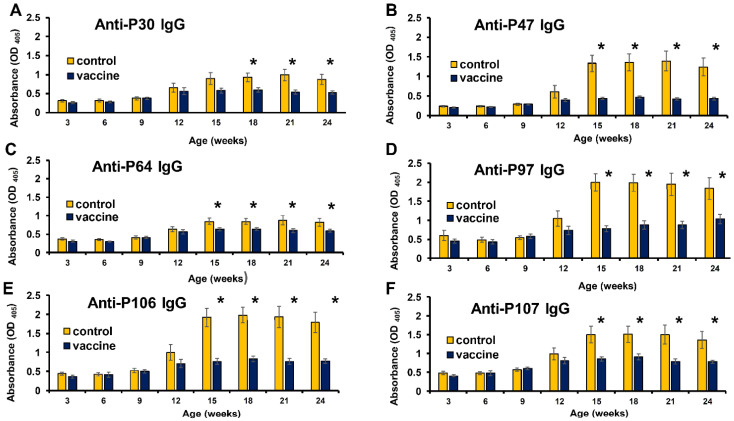
Comparison of anti-C-terminus of the capsid peptide-specific IgGs between two groups of pigs at different ages. Among vaccinated (*n* = 10) and control (*n* = 10) animals, the anti-PCV2 peptide-specific IgGs were detected in the serum samples from pigs at different ages (3, 6, 9, 12, 15, 18, 21, and 24 weeks of age). The C-terminus of the capsid peptides (P30, P47, P64, P97, P106, and P107) were used as coating antigens in the iELISA test. The peptide P30 had a significantly higher binding interaction with pig sera from the control group than with sera from the vaccine group aged 18 weeks to 24 weeks (**A**). Further, the others (P47, P64, P97, P106, and P107) also had a similar effect on pig sera aged 15 weeks to 24 weeks (**B**–**F**). The data were analyzed by using ANOVA and Tukey’s Studentized Range multiple comparisons test using the SAS Enterprise Guide 7.1^®^ software. The data represent the mean ± standard error. Significant *p* values are indicated as * *p* < 0.05.

**Table 1 viruses-14-02373-t001:** Peptide sequences of the truncated carboxyl-terminus (C-terminus) of the capsid protein of PCV2.

Name	PCV Type	Position	Peptide Sequence
P29	2	215–224	VTMYVQFREF
P30	2b	225–233	NLKDPPLNP
P39	2	220–229	QFREFNLKDP
P47	2a	225–233	NLKDPPLKP
P51	2b	222–233	REFNLKDPPLNP
P52	2	225–231	NLKDPPL
P53	2	225–230	NLKDPP
P54	2	226–229	LKDP
P55	2b	230–233	PLNP
P57	2	222–231	REFNLKDPPL
P58	2b	221–232	FREFNLKDPPLN
P59	2b	227–233	KDPPLNP
P62	2b	228–233	DPPLNP
P63	2b	229–233	PPLNP
P64	2a	205–233	CSKYDQDYNIRVTMYVQFREFNLKDPPLKP
P65	2a	228–233	DPPLKP
P66	2b-1c	228–234	DPPLKPK
P67	2d	228–234	DPPLNPK
P68	2d	227–234	KDPPLNPK
P71	2a	227–233	KDPPLKP
P72	2b-1c	227–234	KDPPLKPK
P73	2a	226–233	LKDPPLKP
P74	2b-1c	225–234	NLKDPPLKPK
P77	2a	205–214	SKYDQDYNIR
P78	2a	210–219	DYNIRVTMYV
P79	2	219–228	VQFREFNLKD
P80	2	218–227	YVQFREFNLK
P81	1	219–230	VQFREFILKDP
P82	2	220–227	QFREFNLK
P83	2	217–226	MYVQFREFNL
P84	2a	229–233	PPLKP
P85	2a	229–233	PLKP
P86	2a	205–222	SKYDQDYNIRVTMYVQFR
P88	2	220–226	QFREFNL
P89	2	220–225	QFREFN
P92	1	215–233	RLTIYVQFREFILKDPLNK
P97	2a	220–233	QFREFNLKDPPLKP
P98	2a	223–233	EFNLKDPPLKP
P106	2a	215–233	VTMYVQFREFNLKDPPLKP
P107	2b	215–233	VTMYVQFREFNLKDPPLNP
P108	2a	210–233	DYNIRVTMYVQFREFNLKDPPLKP

Non-conserved amino acids were bold and underlined. P64 was appended with an N-terminal cysteine during synthesis, which was required for conjugation with maleimide-activated carriers [33].

**Table 2 viruses-14-02373-t002:** Peptide sequences of the ORF proteins of PCV2.

Name	PCV Type	Position	Peptide Sequence
C1	2b	ORF2(59–86)	CRTTVKTPSWAVDMMRFNINDFLPPGGGS
C2	2b	ORF2(108–137)	CSPITQGDRGVGSSAVILDDNFVTKATALT
C3	2b	ORF2(195–233)	CHVGLGTAFENSIYDQEYNIRVTMYVQFREFNLKDPPLNP
N1	2b	ORF3(35–66)	CHNDVYISLPITLLHFPAHFQKFSQPAEISDKR
N2	2	ORF6 protein	CMASSTPASPAPSDILSSEPQSERPPGRWT
N3	2	ORF9 protein	MGLGSASSILLAGHVAAEVLPRCCRCRSALVILTAHFFRFQI
P24	2b	ORF2(59–68)	RTTVKTPSWA
P25	2b	ORF2(69–78)	VDMMRFNIND
P26	2b	ORF2(79–86)	FLPPGGGS
P27	2b	ORF2(195–204)	HVGLGTAFEN
P60	2b	ORF2(195–207)	CHVGLGTAFENSIY
P69	2b	ORF3(45–66)	TLLHFPAHFQKFSQPAEISDKR
P70	2b	ORF3(72–92)	CNGHQTPALQQGTHSSRQVTP
P75	2b	ORF3(40–59)	ISLPITLLHFPAHFQKFSQP
P93	2b	ORF2(169–180)	STIDYFQPNNKR
P99	2	ORF2(26–36)	RPWLVHPRHRY
P100	2	ORF2(47–58)	TRLSRTFGYTVK
P101	2	ORF2(132–146)	KATALTYDPYVNYSS
P102	2	ORF2(92–107)	PFEYYRIRKVKVEFWP
P103	2b	ORF2(117–131)	GVGSSAVILDDNFVT
P104	2	ORF2(156–162)	YHSRYFT
P105	2	ORF2(165–185)	PVLDSTIDYFQPNNKRNQLWL
P114	2	ORF10 protein	CMSTAQEGVLTVVALTVYPKVRERRVLKMPFFLLQR
P115	2	ORF11 protein	CMNNKNHYEVIKKTQ
P119	2b	ORF2(220–233)	QFREFNLKDPPLNP
P120	2	ORF7 protein	CMAAGAVSSSAVTPPWIRHS
P121	2	ORF8 protein	CMDIDHTVSVDHPTAASHKSHQ
P122	2a	ORF2(224–232)	NLKDPPLK
P126	2b	ORF3(35–66)	HNDVYISLPITLLHFPAHFQKFSQPAEISDKR
P127	2a	ORF2(51–65)	RTFGYTVKATTVRTP
P128	2b	ORF2(72–85)	MRFNINDFVPPGGG
P129	2a	ORF2(127–137)	DNFVTKATALT
P130	2a	ORF2(166–173)	VLDSTIDY
P131	2a	ORF2(186–191)	RLQTSA

Peptides C1, C3, N1, N2 [42], P60, P114, P115, P120, and P121 were appended with an N-terminal cysteine during synthesis, which was required for conjugation with maleimide-activated carriers.

**Table 3 viruses-14-02373-t003:** Characterization of mAbs against the peptide P64.

mAb	Heavy Chain	Light Chain	ELISA *			IFA ^#^	Minimal Linear Epitope
			PCV2a Peptide P64	PCV2b Peptide C3	VLP	PCV2 FA Substrate Slide	PCV2a/PCV2b Peptide
3H11	IgG2a	κ	+++++	+++++	+	+	PPLKP/DPPLNP
4C9	IgG1	κ	+++++	−	−	+	−
4F6	IgG1	κ	+++++	++++	−	−	QFREFNLK
4H9	IgG1	κ	+++++	−	−	+	−
6E3	IgG2b	κ	+++++	−	−	−	−
6H11	IgG2b	κ	+++++	+++++	++	+	PPLKP/PPLNP

*—means: 0.2 > OD value; + means: 0.7 > OD value > 0.2; ++ means: 1.2 > OD value ≥ 0.7; +++++ means: OD value ≥ 2.2. #—means: the rate of positively stained cells = 0; ± means: ambiguous positive staining; + means: 20% > the rate of positively stained cells > 0; ++ means: 40% > the rate of positively stained cells ≥ 20%; +++ means: 60% > the rate of positively stained cells ≥ 40%; ++++ means: 80% > the rate of positively stained cells ≥ 60%; +++++ means: the rate of positively stained cells ≥ 80%.

## Data Availability

Data sharing does not apply to this article.

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
