# Peer review of "Carboxyl-Terminal Decoy Epitopes in the Capsid Protein of Porcine Circovirus Type 2 Are Immunogenicity-Enhancers That Elicit Predominantly Specific Antibodies in Non-Vaccinated Pigs"

_viruses, 2022, doi:10.3390/v14112373_

Round 1

Reviewer 1 Report

This is an interesting work in that the author used the peptide P64 of PCV2a Cap to immunize mice to generate some anti-P64 mAbs. Using these mAbs, the author found that the 220QFREFNLKDPPLKP233-containing peptides of PCV2a Cap could use as a tool to differentiate vaccinated from PCV2-infected non-vaccinated animals. As the author mentioned that the usage of these peptides still needs further study to figure out if they can also be used for DIVA in the field, because the spreading PCV2 strains are mainly PCV2b and PCV2d.

Author Response

Response to Reviewer 1 Comments

Dear reviewer,

Thank you for allowing me to submit a revised draft of my manuscript ” The Decoy Epitopes Elicit Swine Antibodies by Connecting to the Carboxyl-terminal Epitope of the Capsid Protein of Porcine Circovirus Type 2” (now renamed ” Carboxyl-terminal Decoy Epitopes in the Capsid Protein of Porcine Circovirus Type 2 Are Immunogenicity-enhancers to Elicit Predominantly Specific Antibodies in Non-vaccinated Pigs”) to viruses. I appreciate the time and effort that you have dedicated to providing your valuable feedback on my manuscript. I am grateful to you for your insightful comments on my paper. I have been able to incorporate changes to reflect most of the suggestions provided by the reviewers. I have used the "Track Changes" function in Microsoft Word so that changes are easily visible to you.

Point 1: This is an interesting work in that the author used the peptide P64 of PCV2a Cap to immunize mice to generate some anti-P64 mAbs. Using these mAbs, the author found that the 220QFREFNLKD PPLKP233-containing peptides of PCV2a Cap could use as a tool to differentiate vaccinated from PCV2-infected non-vaccinated animals. As the author mentioned that the usage of these peptides still needs further study to figure out if they can also be used for DIVA in the field, because the spreading PCV2 strains are mainly PCV2b and PCV2d.

Response 1: Thank you for your kind suggestion. According to your suggestion, I added: “but it must be taken into account the major spreading PCV2 strains in the field” in the last sentence of 4.Discussion [This change can be found in lines 671–672]. Further, I address additional information and concepts in the Introduction to let the reader understand current PCV2 immunology [This change can be found in lines 38–72, 82–90, 94, 106–118, and 150–156].

Reviewer 2 Report

This is a topic of great interest due to the high economic impact of PCV2, and that vaccination is the most cost-effective strategy to control the associated diseases. As noted by the author, existing PCV2a-based vaccines are increasingly inadequate due to the shifting prevalence of circulating viral subtype strains. It is essential to identify decoy epitopes that are obstacles to developing new, more broadly-neutralizing vaccines. Numerous studies have previously characterized linear B cell decoy epitopes in PCV2. This work is in concordance with prior findings, and extends our understanding of the precise identity and role of the C-terminal decoy epitope in the immune response to PCV2.

Strengths of this work are:

* Decoy epitopes are mapped with high resolution to particular peptide segments and minimal epitope residues. This is very useful, for example, in that no additional work would be needed to support future attempts at engineering artificial antigens immune-dampened at this site, or to perform in silico analysis of cross-reactivity with other viral strains.

* This work appropriately addresses the native folded structure of the decoy epitopes. As most B cell epitopes are conformational, the 3D structure is a necessary factor to consider in supporting the validity of any results obtained. Potential limitations of working with VLPs are nicely overcome by examination of infected porcine kidney cells.

* The figures present results clearly, with appropriate analyses and statistical consideration for multiple comparisons.

* The introduction and discussion section, along with earlier findings cited, give an accurate and informative picture of how this work fits into an active area of ongoing research.

The Abstract and Introduction additionally need to identify concisely the critical terms and concepts for the reader who is not intimately familiar with current PCV immunology. The rationale, purpose, and value of identifying these epitopes should be made clear. Some key points are that there is a standard vaccine that elicits some immunity, but preferentially restricted to certain strains. Decoy epitopes are immunodominant, but poorly conserved, thereby limiting the breadth of elicited immunity across strains. The authors have identified the molecular target (epitope) of the strain-specific, predominant response to infection. The evidence presented here supports the hypothesis that the C-terminus epitope serves as a strain-specific decoy, such that the immune response does not protect against infection by other strains.

There are editorial improvements that could help the clarity and strength of the message.

The title of this paper is confusing as written. It suggests that decoy epitopes are, themselves, somehow forming connections to the C-terminal epitope.

Some possible examples of how the title could be reworded are as follows.

Carboxyl-terminal Decoy Eptiopes in Porcine Circovirus Type 2 Capsid Protein Elicit Swine Antibodies that are not Broadly Neutralizing Across Strains.

Porcine Circovirus Type 2 Capsid Protein Carboxyl-terminal Decoy Epitopes Elicit Non-Neutralizing Swine Antibodies.

In the abstract:

“The data show two 220QFREFNLKDPPLKP233-containing peptides had significantly higher interactive binding with pig sera from the control group than that sera from the VLP-vaccine group, particularly swine age 15 weeks to 24 weeks, and these pigs had been suffered PCV2 infection.”

could be rewritten as

“The data show two 220QFREFNLKDPPLKP233-containing peptides had significantly higher binding reactivity with sera from PCV2-infected pigs in the control group than with sera from the VLP-vaccine group, particularly seen in sera from swine aged 15 weeks to 24 weeks.”

In section 3.5

“Those C-terminus of the capsid peptides (P30, P47, P65, P71, P73, P84, P97, P98, P106, 430 and P108) had significantly higher interactive binding with pig sera from the control group than those sera from the vaccine group (Figure 4B).”

could be rewritten as

“C-terminal capsid peptides (P30, P47, P65, P71, P73, P84, P97, P98, P106, 430 and P108) had significantly higher binding interaction with pig sera from the control group than with sera from the vaccine group (Figure 4B).”

In the discussion section:

“Since no commercial PCV2b-based vaccines are available to use, then reflected by the change in genotype prevalence to PCV2d, this should be further studied to prove it.”

could be rewritten as

“The shift in circulating genotype prevalence to PCV2d may reflect the lack of available commercial PCV2b-based vaccines, which warrants further study for confirmation.”

Author Response

Response to Reviewer 2 Comments

Dear reviewer,

Thank you for allowing me to submit a revised draft of my manuscript ” The Decoy Epitopes Elicit Swine Antibodies by Connecting to the Carboxyl-terminal Epitope of the Capsid Protein of Porcine Circovirus Type 2” (now renamed ” Carboxyl-terminal Decoy Epitopes in the Capsid Protein of Porcine Circovirus Type 2 Are Immunogenicity-enhancers which Elicit Predominantly Specific Antibodies in Non-vaccinated Pigs”) to viruses. I appreciate the time and effort that you have dedicated to providing your valuable feedback on my manuscript. I am grateful to you for your insightful comments on my paper. I have been able to incorporate changes to reflect most of the suggestions provided by the reviewers. I have used the "Track Changes" function in Microsoft Word so that changes are easily visible to you.

Point 1: The Abstract and Introduction additionally need to identify concisely the critical terms and concepts for the reader who is not intimately familiar with current PCV immunology. The rationale, purpose, and value of identifying these epitopes should be made clear. Some key points are that there is a standard vaccine that elicits some immunity, but preferentially restricted to certain strains. Decoy

epitopes are immunodominant, but poorly conserved, thereby limiting the breadth of elicited immunity across strains. The authors have identified the molecular target (epitope) of the strain-specific, predominant response to infection. The evidence presented here supports the hypothesis that the C-terminus epitope serves as a strain-specific decoy, such that the immune response does not

protect against infection by other strains.

Response 1: Thank you for your kind suggestion. According to your suggestion, I address additional information and concepts in the Abstract and Introduction to let the reader understand current PCV2 immunology [This change can be found in lines 16, 26–29, 38–72, 82–90, 94, 106–118, and 150–156].

Point 2: There are editorial improvements that could help the clarity and strength of the message.

The title of this paper is confusing as written. It suggests that decoy epitopes are, themselves, somehow forming connections to the Cterminal epitope. Some possible examples of how the title could be reworded are as follows. Carboxyl-terminal Decoy Eptiopes in Porcine Circovirus Type 2 Capsid Protein Elicit Swine Antibodies that are not Broadly Neutralizing Across Strains.

Porcine Circovirus Type 2 Capsid Protein Carboxyl-terminal Decoy Epitopes Elicit Non-Neutralizing Swine Antibodies.

Response 2: Thank you for pointing this out. According to your suggestion, I changed the title. Although these epitopes (229PPLKP233 or 228DPPLNP233) were determined as conformational epitopes by mAbs and immunofluorescence assay (PCV2 virus-infected porcine kidney cells), VLP- immunoassays, and the 3D structure analysis, the decoy epitopes (220QFREFNLKDPPLKP233-containing peptides) elicited antibodies which could not produce positive staining in PCV2a-infected cells by immunofluorescence assay (QFREFNLKDPPLKP-inoculated mouse sera did not produce positive staining in PCV2a-infected cells, which was noted in unpublished observations). Considering I have not done the QFREFNLKDPPLKP-inoculated SPF pig sea test, I could not say that “ Decoy Epitopes Elicit Non-Neutralizing Swine Antibodies”. Therefore I reworded the title as ” Carboxyl-terminal Decoy Epitopes in the Capsid Protein of Porcine Circovirus Type 2 Are Immunogenicity-enhancers which Elicit Predominantly Specific Antibodies in Non-vaccinated Pigs [This change can be found in lines 2–5].

Point 3: In the abstract: “The data show two 220QFREFNLKDPPLKP233-containing peptides had significantly higher interactive binding with pig sera from the control group than that sera from the VLP-vaccine group, particularly swine age 15 weeks to 24 weeks, and these pigs had been suffered PCV2 infection.” could be rewritten as “The data show two 220QFREFNLKDPPLKP233-containing peptides had significantly higher binding reactivity with sera from PCV2-infected pigs in the control group than with sera from the VLP vaccine group, particularly seen in sera from swine aged 15 weeks to 24 weeks.”

Response 1: Thank you for your kind suggestion. According to your kind suggestion for editorial improvements, I changed that sentence as your suggestion [This change can be found in lines 26–29 ].

Point 4: In section 3.5: “Those C-terminus of the capsid peptides (P30, P47, P65, P71, P73, P84, P97, P98, P106, 430 and P108) had significantly higher interactive binding with pig sera from the control group than those sera from the vaccine group (Figure 4B).” could be rewritten as “C-terminal capsid peptides (P30, P47, P65, P71, P73, P84, P97, P98, P106, 430 and P108) had significantly higher binding interaction with pig sera from the control group than with sera from

the vaccine group (Figure 4B).”

Response 1: Thank you for your kind suggestion. According to your kind suggestion for editorial improvements, I changed that sentence as your suggestion [This change can be found in lines 514–516].

Point 5: In the discussion section: “Since no commercial PCV2b-based vaccines are available to use,

then reflected by the change in genotype prevalence to PCV2d, this should be further studied to prove it.” could be rewritten as “The shift in circulating genotype prevalence to PCV2d may reflect the lack of available commercial PCV2b-based vaccines, which warrants further study for confirmation.”

Response 5: Agree. According to your kind comments for editorial improvements, I changed that sentence as per your suggestion [This change can be found in lines 597–599].